# Clinical Significance of Cytoplasmic IgE-Positive Mast Cells in Eosinophilic Chronic Rhinosinusitis

**DOI:** 10.3390/ijms21051843

**Published:** 2020-03-07

**Authors:** Yuka Gion, Mitsuhiro Okano, Takahisa Koyama, Tokie Oura, Asami Nishikori, Yorihisa Orita, Tomoyasu Tachibana, Hidenori Marunaka, Takuma Makino, Kazunori Nishizaki, Yasuharu Sato

**Affiliations:** 1Division of Pathophysiology, Okayama University Graduate School of Health Sciences, Okayama 700-8558, Japan; gion@okayama-u.ac.jp (Y.G.); pe0r79kv@s.okayama-u.ac.jp (T.O.); asami.kei@s.okayama-u.ac.jp (A.N.); 2Department of Pathology, Okayama University Graduate School of Medicine, Dentistry and Pharmaceutical Sciences, Okayama 700-8558, Japan; 3Department of Otolaryngology of Head and Neck Surgery, Okayama University Graduate School of Medicine, Dentistry and Pharmaceutical Sciences, Okayama 700-8558, Japan.; koyatakaco@gmail.com (T.K.); marunaka@okayama-u.ac.jp (H.M.); takmak0617@yahoo.co.jp (T.M.); nishizak@cc.okayama-u.ac.jp (K.N.); 4Department of Otorhinolaryngology, International University of Health and Welfare Graduate School of Medicine, Narita 286-8686, Japan; 5Department of Otolaryngology, Head and Neck Surgery, Kumamoto University Graduate School, Kumamoto 860-8556, Japan; y.orita@live.jp; 6Department of Otolaryngology, Japanese Red Cross Society Himeji Hospital, Himeji 670-8540, Japan; tomoyasutachibana@hotmail.co.jp

**Keywords:** mast cell, eosinophilic chronic rhinosinusitis, c-kit, IgE

## Abstract

Cross-linking of antigen-specific IgE bound to the high-affinity IgE receptor (FcεRI) on the surface of mast cells with multivalent antigens results in the release of mediators and development of type 2 inflammation. FcεRI expression and IgE synthesis are, therefore, critical for type 2 inflammatory disease development. In an attempt to clarify the relationship between eosinophilic chronic rhinosinusitis (ECRS) and mast cell infiltration, we analyzed mast cell infiltration at lesion sites and determined its clinical significance. Mast cells are positive for c-kit, and IgE in uncinated tissues (UT) and nasal polyps (NP) were examined by immunohistochemistry. The number of positive cells and clinicopathological factors were analyzed. Patients with ECRS exhibited high levels of total IgE serum levels and elevated peripheral blood eosinophil ratios. As a result, the number of mast cells with membranes positive for c-kit and IgE increased significantly in lesions forming NP. Therefore, we classified IgE-positive mast cells into two groups: membrane IgE-positive cells and cytoplasmic IgE-positive cells. The amount of membrane IgE-positive mast cells was significantly increased in moderate ECRS. A positive correlation was found between the membrane IgE-positive cells and the radiological severity score, the ratio of eosinophils, and the total serum IgE level. The number of cytoplasmic IgE-positive mast cells was significantly increased in moderate and severe ECRS. A positive correlation was observed between the cytoplasmic IgE-positive cells and the radiological severity score, the ratio of eosinophils in the blood, and the total IgE level. These results suggest that the process of mast cell internalization of antigens via the IgE receptor is involved in ECRS pathogenesis.

## 1. Introduction

Chronic rhinosinusitis (CRS) is a condition characterized by chronic mucosal inflammation in paranasal sinuses for more than 12 weeks, which ultimately causes nasal obstruction, rhinorrhea, and posterior nasal discharge. CRS is a disease with various etiologies and pathologies and is often accompanied by headaches and olfactory disorders. One type of CRS is refractory eosinophilic chronic rhinosinusitis (ECRS). Many patients with non-ECRS are cured by antibiotic administration or surgery. However, ECRS forms multiple nasal polyps (NP) in a bilateral nature and tends to relapse even after surgery [1,2]. This disease is relieved by steroid administration, and steroid therapy is considered to be the most effective treatment [3,4]. ECRS lesions are characterized by an infiltration of numerous eosinophils. Furthermore, in these patients, increases in peripheral blood allergic factors are observed, including an increase in eosinophils and IgE. Additionally, this disease frequently occurs in patients with aspirin intolerance, in which aspirin causes asthma [5].

Mast cells have a wide variety of functional molecules expressed on their surface, which facilitate cell activation via chemical mediators that synthesize and secrete various cytokines during allergic or innate immune responses, both of which are heavily influenced by the involvement of the acquired immune system [6,7,8]. Mast cells have also been shown to interact with immune cells, such as T-cells, B-cells, and dendritic cells, and to participate in transplant rejection and tumor immunity [9,10] while also assisting in protection against viral and bacterial infection via Toll-like receptors (TLRs) [11,12]. Regarding allergies, it has been confirmed that mast cells interact with B-cells in the nasal mucosa and bronchial mucosa at the time of allergic inflammation leading to the promotion of local IgE production, suggesting the existence of an allergic exacerbation cycle [13]. Further, mast cells activated by local inflammation express CD40L (CD154) and produce Th2 cytokines such as interleukin (IL)-4 and IL-13. It has also been shown that IgE production is induced by co-culturing nasal mucosal mast cells from allergic rhinitis patients with B-cells in vitro [13]. This IgE produced in inflamed regions serves to enhance high-affinity IgE receptor (FcεRI) expression on the surface of mast cells and markedly increases sensitivity to this antigen [14]. Thus, it seems that the allergic exacerbation cycle, responsible for further promoting the inflammatory response, will be initiated in the local area of allergic inflammation. Herein, we focused on the fact that many patients with ECRS had a predisposition to allergies, and thus, the current study sought to investigate mast cell expression and its clinical significance in ECRS lesions.

## 2. Results

### 2.1. Patients’ Characteristics

Clinical features are shown in Table 1. Prior to surgery, each CRS patient was examined for serum levels of IgE, eosinophil ratio, and 1-s forced expiratory volume/forced vital capacity (FEV_1_/FVC) ratio. Serum IgE levels were found to be high in severe cases of ECRS (median; 476.2 IU/mL, *n* = 10), while moderate and severe ECRS cases showed an increased eosinophil count in the peripheral blood. Specifically, the mean blood eosinophil ratio was 5.6% for moderate ECRS and 10.8% for severe ECRS. Further, the mean cognitive threshold in the baseline olfactory examination was particularly high in patients with ECRS. Among 71 CRS patients, 57 patients exhibited nasal polyps (CRSwNP), with the remainder exhibiting no visible NP in the middle meatus (CRSsNP; *n* = 14). Samples were divided into six groups according to the CRS phenotype: uncinated process tissues (UT) from non-CRS (*n* = 13), UT from CRSsNP (*n* = 14), NP from non-ECRS (*n* = 27), NP from mild ECRS (*n* = 8), NP from moderate ECRS (*n* = 12), and NP from severe ECRS (*n* = 10) (Figure 1).

### 2.2. Histological Evaluation and Pathophysiological Significance of c-Kit-Positive Cells

The c-kit-positive mast cells were observed in each group (Figure 2), with the number of c-kit-positive cells in UT and NP ranging from 0–18 (median: 5.3 cells/HPF) and from 2–36 (median: 9.6 cells/HPF), respectively, per high-power field (HPF). The number of c-kit-positive cells was significantly higher in NP than in UT (*p* < 0.001; Figure 3A), which was then compared between the six study groups. No significant differences were observed between the UT from the non-CRS group and the CRSsNP group; however, the c-kit-positive cells were significantly increased in the groups forming NP as compared to CRSsNP. The Kruskal–Wallis test revealed a significant difference in the number of infiltrating c-kit-positive cells among these groups (*p* < 0.0001). However, no significant difference was observed between non-ECRS and each ECRS group. In addition, the severity of ECRS and the number of c-kit-positive cells were not directly related.

Next, we pathophysiologically characterized the degree of c-kit-positive cell infiltration in sinonasal tissues. A significant positive correlation was found between the number of infiltrating c-kit-positive cells and the radiological severity of CRS (r = 0.309, *p* = 0.006; Figure 3B). Conversely, no correlation was observed between the number of infiltrating c-kit-positive cells and peripheral blood eosinophilia (r = 0.123, *p* = 0.265; Figure 3C) or total serum IgE level (r = 0.043, *p* = 0.710; Figure 3D).

### 2.3. Histological Evaluation and Pathophysiological Significance of IgE-Positive Cells

IgE-positive mast cells were observed in each group (Figure 4) and classified into two types: mast cells that showed IgE-positive only in the membrane were termed “membrane IgE-positive mast cells”, while mast cells with IgE positivity in the cytoplasm were designated “cytoplasmic IgE-positive mast cells”. The number of each cell type was enumerated.

A higher number of membrane IgE-positive mast cells was observed in cases with NP compared to those without, with the numbers of membrane IgE-positive mast cells in the UT and NP ranging from 0–16 (median: 5.2 cells/HPF) and from 0–27 (median: 9.5 cells/HPF), respectively, per HPF. The Kruskal–Wallis test further demonstrated that, although the number was not different between UT from non-CRS and UT from CRSsNP, membrane IgE-positive mast cells increased in NP-forming groups. In addition, moderate ECRS showed a statistically significant increase compared to non-CRS and CRSsNP (UT from non-CRS: *p* < 0.05; UT from non-ECRS: *p* < 0.05) (Figure 5A). However, no significant difference was observed between non-ECRS and ECRS. Furthermore, comparing the clinical data with membrane IgE-positive cells revealed that the number of membrane IgE-positive mast cells was also positively correlated with radiological severity of CRS (r = 0.470, *p* < 0.001; Figure 5B), peripheral blood eosinophilia (r = 0.327, *p* = 0.002; Figure 5C), and the total serum IgE level (r = 0.458, *p* < 0.001; Figure 5D).

Furthermore, cytoplasmic IgE-positive mast cells were found in lesions of moderate and severe ECRS (Figure 6A–F). The number of these cells was statistically higher in moderate and severe ECRS than in the other groups (*p* < 0.05 and *p* < 0.01, respectively; Figure 6G). The number of cytoplasmic IgE-positive mast cells was also significantly correlated to radiological severity of CRS (r = 0.280, *p* = 0.012; Figure 6H), peripheral blood eosinophilia (r = 0.404, *p* < 0.001; Figure 6I), and total serum IgE (r = 0.500, *p* < 0.001; Figure 6J).

## 3. Discussion

In this study, an increase in mast cells was observed in non-ECRS and ECRS lesions. Two types of human mast cells have been described: mast cells that express both tryptase and chymase, found in the subcutaneous connective tissue (TC-type mast cells; MC_TC_), and mast cells that express only tryptase, found in the airway mucosa and intestinal mucosa (T-type mast cells; MC_T_) [15]. Baba et al. reported increased numbers of MC_TC_ in the epithelium, glands, and submucosa of ECRS polyps, as well as increased numbers of MC_T_ in the glands and submucosa of non-ECRS polyps [16]. Furthermore, it was suggested that the distribution of IgE-positive mast cell subtypes differs between ECRS and non-ECRS [16]. As the present study did not analyze production of tryptase and chymase, it is unclear which type of mast cells increased. However, it is presumed that MC_T_ increased with non-ECRS and MC_TC_ increased with ECRS. In our study, the number of c-kit-positive cells and the number of membrane IgE-positive mast cells increased in cases where NP were formed. In addition, moderate and severe ECRS exhibited an increase in the number of cytoplasmic IgE-positive mast cells. Mast cell endocytosis is caused by strong antigen stimulation and is a phenomenon in which FcεRI and IgE antibodies on the membrane surface are taken into the cytoplasm. Mast cell endocytosis is considered to be the cause of positive cytoplasmic IgE. Increased levels of IgE are believed to increase the amount of FcεRI, which in turn induces increased levels of endocytosis [17]. In patients with ECRS, IgE-positive cells are increased due to an allergic predisposition. In severe cases of ECRS, the majority of patients have asthma, and excessive endocytosis may further activate the allergic reaction. In addition, sustained mucosal inflammation causes the cells to be susceptible to infection from bacteria and viruses, resulting in strong antigen stimulation and further activation of the immune response. The accumulation of these factors results in the activation of mast cells, which further increases IgE levels. As such, we speculate that moderate or severe ECRS lesions may result in an increase in mast cells positive for cytoplasmic IgE. In the present study, cytoplasmic IgE-positive mast cells were significantly increased in moderate and severe ECRS, suggesting that the status of ECRS is primarily allergic, involving activation of mast cells.

As an internal immune mechanism, mast cells are activated upon antigen stimulation to release cytokines, such as IL-4, IL-13, and IL-5, which are classified as type 2 cytokines and are involved in the production of IgE as well as the differentiation/proliferation of eosinophils, thus promoting eosinophil expansion and production of IgE antibody by plasma cells. With increased severity of ECRS, mucosal rupture may continuously occur, resulting in enhanced and sustained antigenic stimulation and further mast cell activation, which subsequently cause an enhanced expression of FcεRI. Propagation of this cycle is considered to lead to endocytosis of mast cells. In this study, since the patients’ blood samples were not preserved due to the nature of the study being retrospective, it was not possible to quantify Th2 cytokines in the blood. In addition, the quantification of these cytokines in tissues was not possible due to the small amount of FFPE samples.

In addition, it has been reported that staphylococcal enterotoxin is likely to be involved in eosinophilic inflammation of the nasal mucosa as a superantigen or as an adjuvant [18]. The staphylococcal enterotoxin of *Staphylococcus aureus* has been detected in NP and mucins, and specific IgE against *S. aureus* enterotoxins has also been detected. Furthermore, the polyclonal increase of nonspecific IgE in the nasal cavity is considered to be important for nasal polyp formation [19], while also inducing differentiation of precursor cells to eosinophils and contributing to proliferation. Furthermore, it has been reported that anti-IgE antibody therapy is an effective treatment for ECRS accompanied by NP [20,21,22]. It has also been reported that anti-IL-5 antibodies were effective against eosinophil-dominated NP [23]. IL-5 is an important cytokine for eosinophil migration and activation. Taken together, it has been suggested that mast cells are involved in the mechanism of ECRS accompanied by the formation of NP.

In a recent study, it has been reported that IgG4-positive cells increase significantly in severe ECRS [24]. Particularly in allergic diseases, IgG4 is considered to act as a blocking agent for IgE antibodies induced by the antigen. Furthermore, mast cells positive for cytoplasmic IgE have been reported in IgG4-related disease (IgG4-RD) [17]. In addition, it has been shown that the number of endocytic mast cells also increases in the lesions of IgG4-RD [17]. Severe cases of IgG4-RD often have multiorgan lesions. Similarly, multiple organ lesions, such as eosinophilic esophagitis, have been reported in severe cases of ECRS [25,26]. While the precise pathogenesis of these diseases is unclear, there may be a link to family predisposition and allergic diseases [27,28].

In conclusion, it is suggested that mast cells that receive antigen stimulation are involved in the pathogenesis of ECRS.

## 4. Materials and Methods

### 4.1. Patients

Seventy-one Japanese patients with CRS (47 males and 24 females; mean age, 55.8 years) were enrolled. Among these, 57 patients exhibited NP, and the remainder demonstrated no visible NP in the middle meatus (*n* = 14) [24]. Patients with CRSwNP were divided into non-ECRS (*n* = 27) and ECRS (*n* = 30) groups based on the Japanese Epidemiological Survey of Refractory Eosinophilic Chronic Rhinosinusitis (JESREC) criteria [5]. In brief, the JESREC scoring system assesses whether it is a unilateral or bilateral disease, the presence of NP, degree of blood eosinophilia, and the dominant shadow of ethmoid sinus in computed tomography (CT) scans. Herein, a case was diagnosed as ECRS if it showed a JESREC score of ≥ 1, and tissue eosinophilia ≥ 70 per HPF; ×400). The severity of ECRS was further determined by the JESREC algorithm using factor A (presence of both blood eosinophilia equal to or greater than 5% and an ethmoid-dominant shadow on a CT scan) and factor B (comorbid bronchial asthma or nonsteroidal anti-inflammatory drug intolerance) as follows: cases negative for both factor A and B, cases positive for either factor A or B, or cases positive for both factor A and B were grouped into mild, moderate, or severe ECRS groups, respectively [5]. Using this algorithm, 30 ECRS patients were categorized as mild (*n* = 8), moderate (*n* = 12), or severe (*n* = 10) ECRS subgroups. All CRSsNP patients were diagnosed as being non-ECRS using these criteria. During surgery, NP and UT were taken from patients with CRSwNP and CRSsNP, respectively. Serum samples were collected from 17 patients (non-ECRS: *n* = 6; mild ECRS: *n* = 4; moderate ECRS: *n* = 2; severe ECRS: *n* = 5). As the control, 13 non-CRS patients (e.g., patients with blowout fractures, posterior ethmoidal cysts, or sphenoidal cysts) with normal UT at inspection were enrolled (four males and nine females; mean age, 61.4 years). After surgery, CRS patients received medications, including macrolides and mucolytic agents for 2 months, together with saline douching, which was continued as long as they could. In addition, NP patients received systemic corticosteroids (prednisolone: started with 20 mg/day, then gradually decreased over 1 month) followed by intranasal corticosteroids. Furthermore, ECRS patients standardly received oral antileukotrienes.

Informed consent for participation in the study was obtained from each patient, and the study was approved by the Human Research Committee of the Okayama University Graduate School of Medicine and Dentistry (reference number 1505-030).

### 4.2. Histological Examination and Immunohistochemistry

All samples used in this study were surgically resected specimens. The surgically removed tissues were fixed in 10% formaldehyde and embedded in paraffin. Serial 3-μm-thick sections were cut from the blocks and stained with hematoxylin and eosin (H&E); the sections were immunohistochemically stained using an automated BOND III stainer (Leica Biosystems, Wetzlar, Germany). Primary antibodies against the following antigens were used: c-kit (diluted antibody; Nichirei Biosciences Inc., Tokyo, Japan) and IgE (1:200; DAKO, Glostrup, Denmark).

### 4.3. Histological Evaluation of c-Kit and IgE-Positive Cell

Mast cells are defined as cells that express both high-affinity IgE receptors and the stem cell factor receptor, c-kit; therefore, we analyzed the number of mast cells by immunostaining for these markers. As dendritic cells also express FcεRI, they also stain positive for IgE immunohistochemistry; however, their shape is significantly different from that of mast cells, allowing us to only quantify cells that could be distinguished as mast cells.

Cell counts were performed by two independent researchers, and the average number was calculated. The cells were counted in three hotspot fields with an HPF (×400), and the average per field was determined. In the hotspot area of c-kit-positive cells, the number of c-kit-positive cells was counted for each specimen. In the hotspot area of IgE-positive cells, the numbers of cells showing IgE-positive in the membrane and cytoplasm were counted separately.

### 4.4. Statistical Analysis

Values are given as the median. The nonparametric Mann-Whitney U test was used to compare data between groups, and Wilcoxon’s signed-rank test was used to analyze data within each group. A Kruskal–Wallis test, followed by a Dunn test, was used for multiple comparisons. Correlation analyses were performed using Spearman’s rank correlation. Statistical analyses were performed with GraphPad Prism 6 software (GraphPad Software, Inc., La Jolla, CA, USA). The *p*-values for sensitivity and specificity were calculated using JMP Pro 13.2 (SAS Institute Inc., Cary, NC, USA), and logistic regression analyses were conducted using STATA 12.1 (StataCorp, College Station, TX, USA). A *p* value less than 0.05 (two-tailed) was considered to be statistically significant.

To compare the c-kit-positive cells and IgE-positive cells among eosinophilic sinusitis, nonsinusitis, chronic sinusitis, and noneosinophilic sinusitis, the Kruskal–Wallis test method, using IBM-SPSS statistics software (version 24; IBM, Armonk, NY, USA), was applied. A *p* value < 0.05 was considered statistically significant.

## Figures and Tables

**Figure 1 ijms-21-01843-f001:**
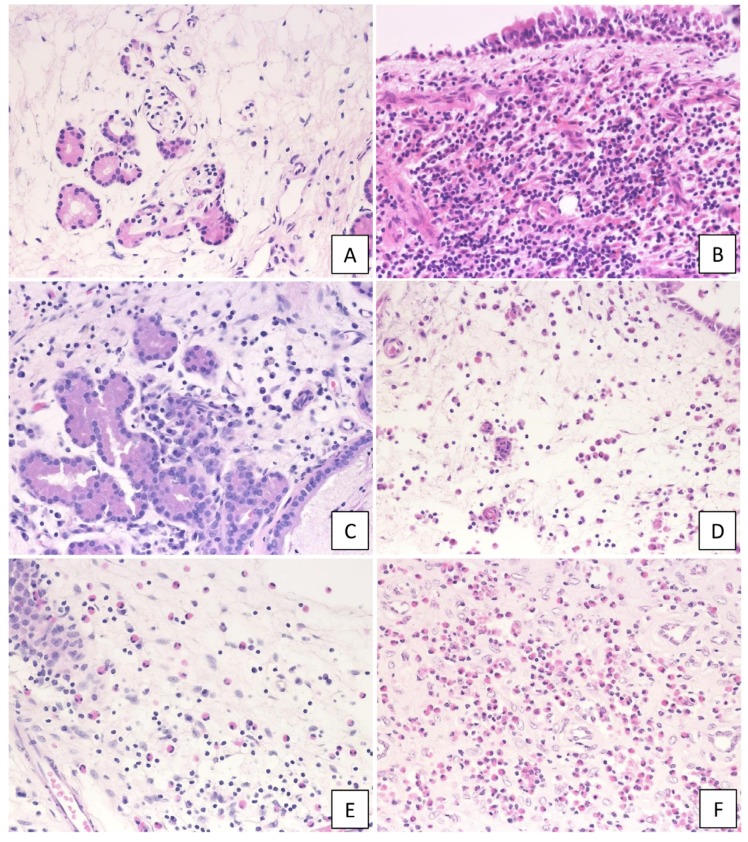
Histological findings of non-chronic rhinosinusitis (CRS), CRS, and eosinophilic chronic rhinosinusitis (ECRS). Hematoxylin and eosin staining at 400× magnification (high-powered field, HPF). (**A**) non-CRS, (**B**) CRS with no visible NP in the middle meatus (CRSsNP), (**C**) non-ECRS, (**D**) mild ECRS, (**E**) moderate ECRS, and (F) severe ECRS. In ECRS, infiltration of eosinophils under the mucosa was observed.

**Figure 2 ijms-21-01843-f002:**
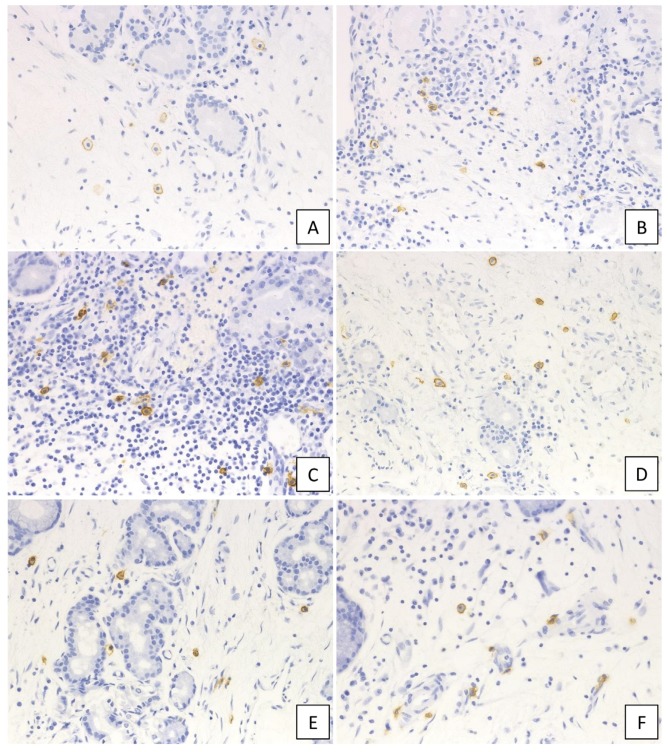
Immunohistochemical staining of c-kit in the diseased tissue. (**A**) non-CRS, (**B**) CRSsNP, (**C**) non-ECRS, (**D**) mild ECRS, (**E**) moderate ECRS, and (**F**) severe ECRS. The surface membrane of mast cells was positive for c-kit (400× magnification).

**Figure 3 ijms-21-01843-f003:**
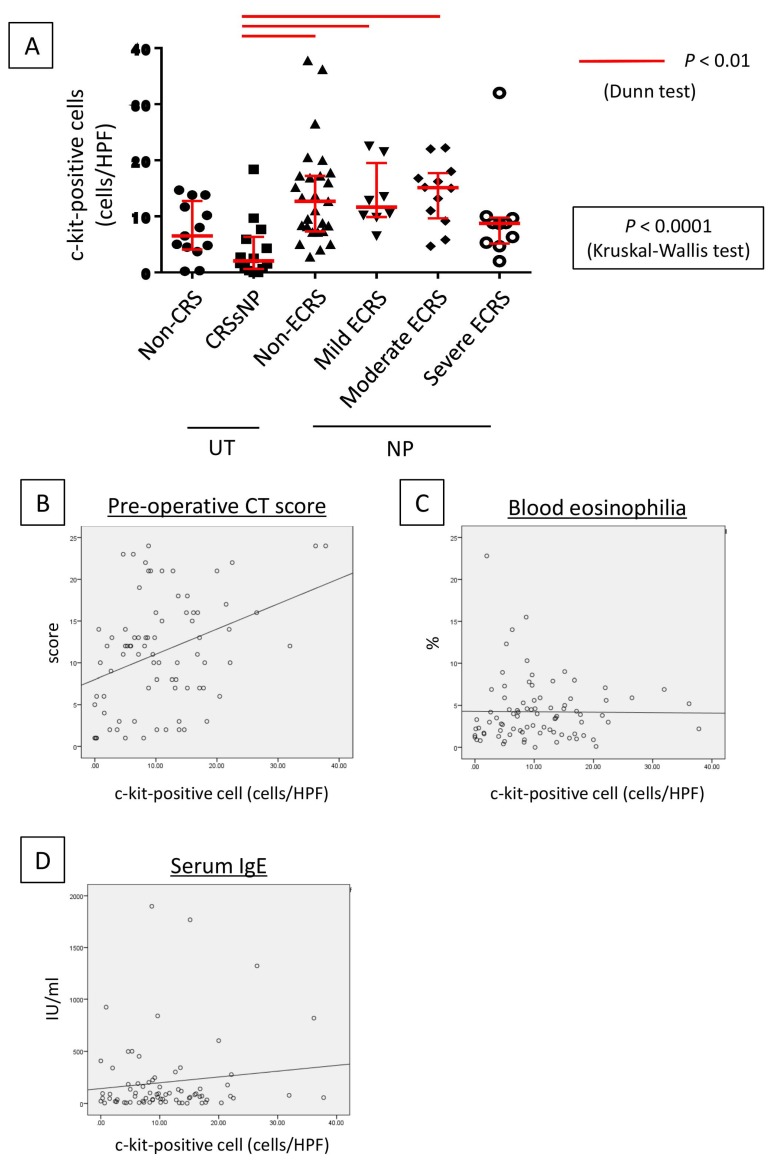
Lesion infiltration by c-kit-positive cells and the number of c-kit-positive cells. (**A**) Examination of the number of c-kit positive cells in the hotspot of each case. The number of c-kit-positive cells increased in the nasal polyp groups (*p* < 0.0001, Kruskal–Wallis test). Relationship between the number of c-kit positive cells and the preoperative CT score (**B**), eosinophil ratio in peripheral blood (**C**), and total serum IgE level (**D**).

**Figure 4 ijms-21-01843-f004:**
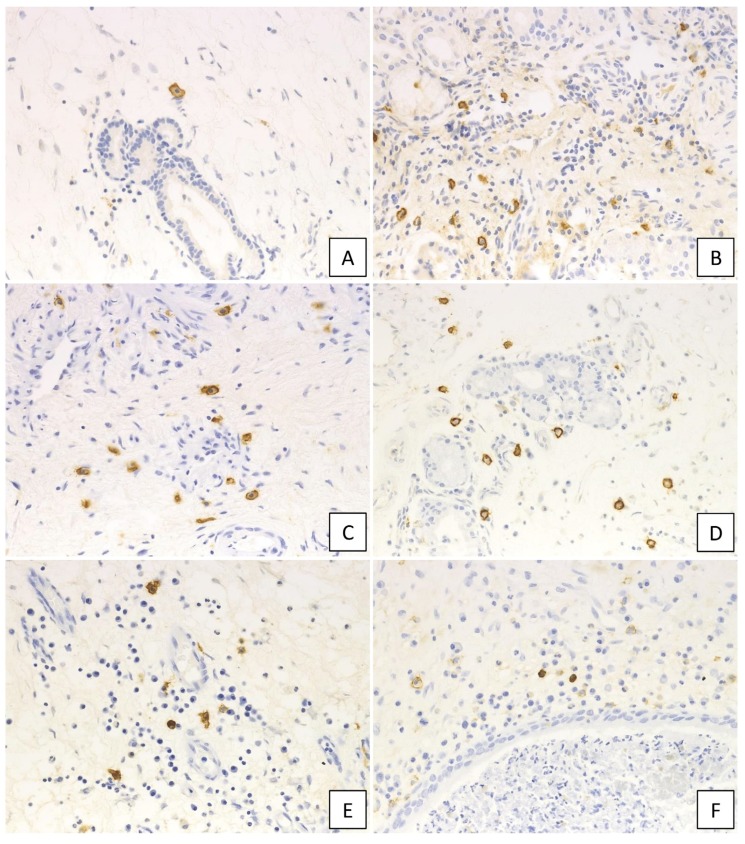
Immunohistochemical findings of IgE in the diseased tissue. (**A**) non-CRS, (**B**) CRSsNP, (**C**) non-ECRS, (**D**) mild ECRS, (**E**) moderate ECRS, and (**F**) severe ECRS. IgE-positive cells increased in the diseased tissue (400× magnification).

**Figure 5 ijms-21-01843-f005:**
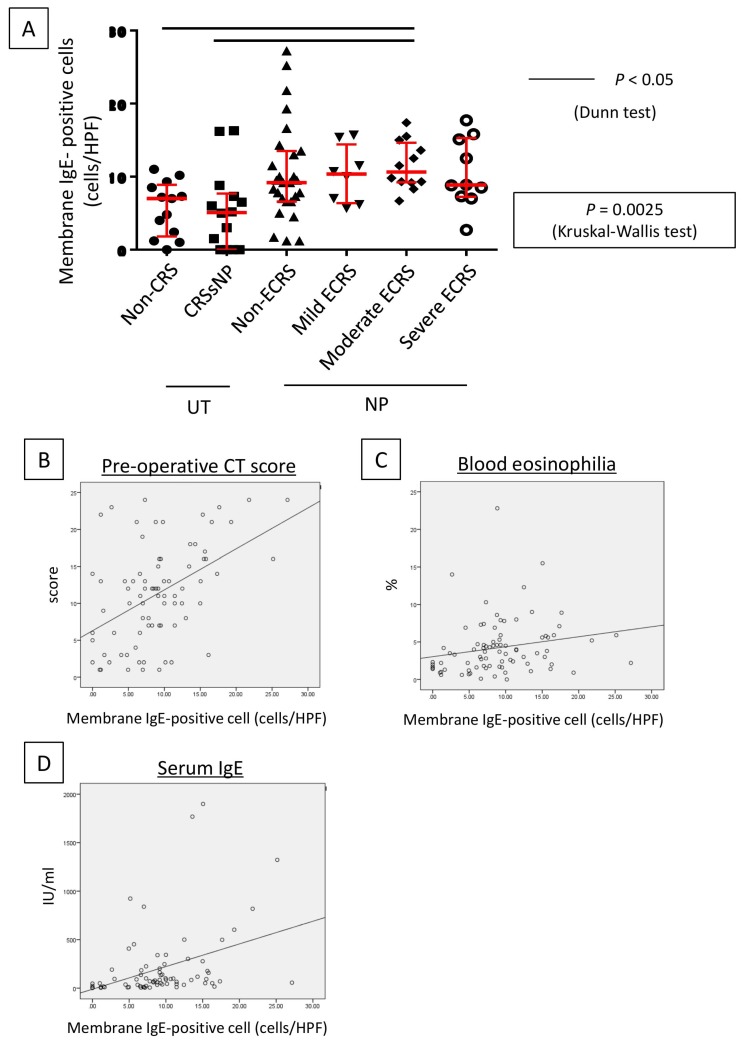
Lesion infiltration by membrane IgE-positive cells and the number of IgE-positive cells. (**A**) Determination of membrane IgE-positive cell number in the hotspot of each case. The number of membrane IgE-positive cells increased in the nasal polyp groups (*p* < 0.0025, Kruskal–Wallis test). Relationships between the number of membrane IgE-positive cells and the preoperative CT score (**B**), eosinophil ratio in peripheral blood (**C**), and total serum IgE level (**D**).

**Figure 6 ijms-21-01843-f006:**
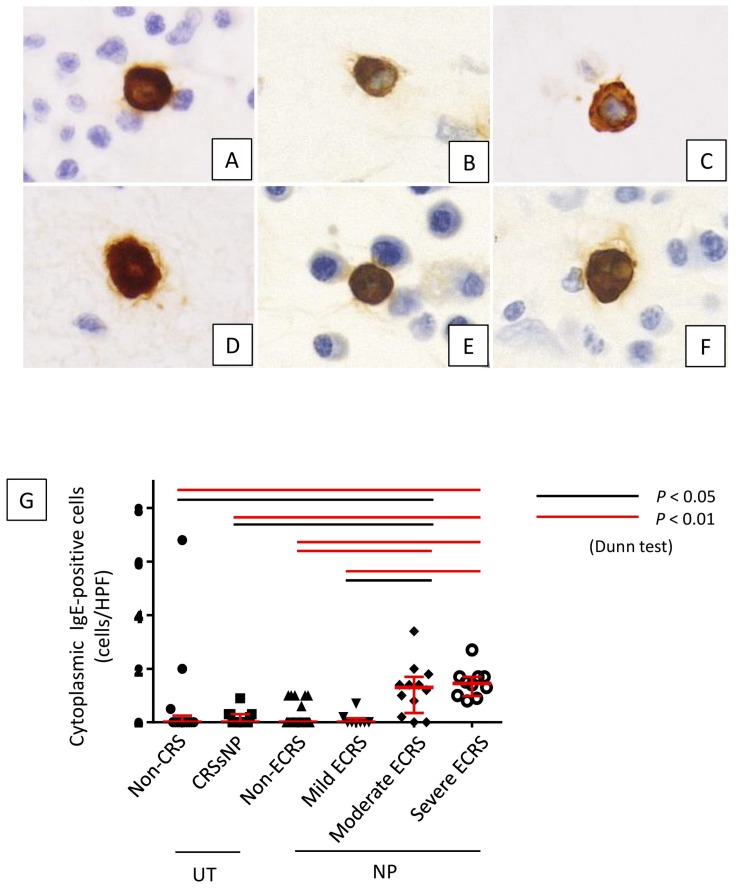
Lesion infiltration by cytoplasmic IgE-positive cells in each lesion. (**A**) non-CRS, (**B**) CRSsNP, (**C**) non-ECRS, (**D**) mild ECRS, (**E**) moderate ECRS, and (**F**) severe ECRS. (**G**) Cytoplasmic IgE-positive mast cells were significantly increased in moderate and severe ECRS lesions (*p* < 0.05 and *p* < 0.01, respectively). Relationships between the number of cytoplasmic IgE-positive cells and the preoperative CT score (**H**), eosinophil ratio in peripheral blood (**I**), and total serum IgE level (**J**).

**Table 1 ijms-21-01843-t001:** Subjects’ characteristics.

Groups	Non-CRS (UT)	CRSsNP (UT)	Non-ECRS (NP)	Mild ECRS (NP)	Moderate ECRS (NP)	Severe ECRS (NP)
Number	13	14	27	8	12	10
Age (years old)	61.4 (41–92)	60.5 (35–75)	57.6 (36–84)	56.8 (34–70)	52.5 (32–78)	49.5 (33–74)
Sex (female/male)	9/4	6/8	8/19	2/6	5/7	3/7
Blood eosinophil rate (%)	2.1 (0–5.9)	2.5 (0.8–8.6)	3.2 (0.1–7.3)	3.8 (2.6–4.7)	5.6 (1.7–7.9)	10.8 (4.6–22.8)
Serum total IgE (IU/mL)	34.5 (2–99)	141.7 (4–923)	164.6 (4–1322)	150.4 (10–452)	256.4 (34–1768)	476.2 (34–1899)
FEV1/FVC ratio (%)	83.8 (68.7–92.7)	77.3 (66.5–84.7)	78.3 (47.2–91.9)	76.3 (73.2–86.4)	76.0 (49.2–92.3)	72.8 (52.6–89.5)
CT grading score (Lund–Mackay)	1.9 (1–3)	6.5 (1–14)	13.4 (3–24)	14.3 (8–22)	13.4 (7–21)	16.6 (10–24)
Comorbidity of asthma (*n*)	0	0	5	0	5	10
Comorbidity of NSAIDs intolerance (*n*)	0	0	2	0	1	3

CRS, chronic rhinosinusitis; ECRS, eosinophilic chronic rhinosinusitis; CRSsNP, chronic rhinosinusitis without nasal polyps; UT, uncinated tissues; NP, nasal polyps; FEV1/FVC ratio, 1-s forced expiratory volume/forced vital capacity ratio; NSAIDs, nonsteroidal anti-inflammatory drugs.

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
