# Peer review of "Clinical Significance of Cytoplasmic IgE-Positive Mast Cells in Eosinophilic Chronic Rhinosinusitis"

_ijms, 2020, doi:10.3390/ijms21051843_

Round 1
Reviewer 1 Report
The authors have performed some but not all recommendations made by the Reviewers. There are remaining issues that if addressed will significantly enhance the validity of the proposed findings:
The authors should performed a double immunostaining experiment using a dendritic cell marker to demonstrate the specificity of the IgE staining on mast cells. This is an important control as dendritic cells may also exhibit surface IgE levels. The levels of Th2 cytokines should be measured in the serum and correlations should be made with the authors findings on mast cells. The discussion needs to be shortened and focused on describing the implications of the authors findings.
Author Response
Reviewer 1
The authors have performed some but not all recommendations made by the Reviewers. There are remaining issues that if addressed will significantly enhance the validity of the proposed findings:
The authors should performed a double immunostaining experiment using a dendritic cell marker to demonstrate the specificity of the IgE staining on mast cells. This is an important control as dendritic cells may also exhibit surface IgE levels. The levels of Th2 cytokines should be measured in the serum and correlations should be made with the authors findings on mast cells. The discussion needs to be shortened and focused on describing the implications of the authors findings.
[Reply]
Thank you for your comments.
In this study, double immunostaining could not be examined. We have analyzed mast cells in IgG4-related diseases in the previous study (references 17), in the result indicate that mast cells showed double positive for c-kit and IgE, and also morphologically cytoplasmic IgE-positive cells were consistent with mast cell and quite different from dendritic cells. These findings were checked by hematopathologist.
The modified part was displayed in blue in the text (pp.15, lines 2-5).
In addition, as you point out, we thought that quantification of Th2 cytokines was necessary and we tried to quantify Th2 cytokines in FFPE tissue of lesions using real-time PCR. However, we were unable to obtain high quality results due to the small amount of tissues. This study was a retrospective study and could not analyze Th2 cytokine levels in the serum because samples of patients’ blood were not preserved.
The modified part was displayed in blue in the text (pp.10, lines 12-15).
Reviewer 2 Report
In this revised manuscript, the authors made some changes and slightly improved the manuscript. However, the main point of the paper was that increased mast cells, particularly those cytoplasmic IgE-positive cells, are important for the pathogenesis of ECRS but the data were not very convincing.
In response to a raised concern of insufficient data in the previous review, the authors split some of the figures in the previous manuscript to increase the number of total figures. This did not make a meaningful difference from last time.
Figure 3A. There were three comparisons, but all with CRSsNP. There was no comparison between Non-ECRS and Mild, Moderate and Severe ECRS groups?
Figure 5A. Similar issues with this figure. There was no comparison between Non-ECRS group with other ECRS groups. It seems that there was no difference between these groups.
Author Response
Reviewer 2
In this revised manuscript, the authors made some changes and slightly improved the manuscript. However, the main point of the paper was that increased mast cells, particularly those cytoplasmic IgE-positive cells, are important for the pathogenesis of ECRS but the data were not very convincing.
Figure 3A. There were three comparisons, but all with CRSsNP. There was no comparison between Non-ECRS and Mild, Moderate and Severe ECRS groups?
[Reply]
Thank you for your comments.
This study examined the number of positive cells in six groups (non-CRS, CRSsNP, non-ECRS, mild ECRS, moderate ECRS and severe ECRS). The c-kit-positive cells showed a significant increase in the nasal polyps formation group as compared to CRSsNP. However, as a result of our analysis, no correlation was found between the severity of ECRS and the number of c-kit-positive mast cells. The last figure change examined the correlation between clinical data and the number of positive cells. As a result, there were positive correlation between the number of c-kit-positive cells and the radiological severity of CRS. It is shown in the figure only when a significant difference was observed.
The modified parts were added in blue in the text (pp.6, lines 8-11, 13-15).
Figure 5A. Similar issues with this figure. There was no comparison between Non-ECRS group with other ECRS groups. It seems that there was no difference between these groups.
[Reply]
Thank you for your comments.
No significant differences were found in membrane IgE-positive mast cells in non-ECRS. However cytoplasmic IgE-positive mast cells showed a significant increase in moderate and severe ECRS. The IgE positivity in the cytoplasm indicates that mast cells were causing endocytosis. These findings suggest that mast cell activation may be involved in moderate or severe ECRS.
We have modified the text (pp.8, lines 1-3). The modified parts were added in blue in the text.
Reviewer 3 Report
The authors revealed the importance of mast cells in eosinophilic chronic rhinosinusitis. They found that the number of mast cells with membranes positive for c-kit and IgE increased significantly in lesions forming nasal polyp. In addition, they found that the number of cytoplasmic IgE-positive mast cells was significantly increased in moderate and severe ECRS. In general, manuscript is well organized and well written. There are a few changes which would improve the content.
1: Did the number of mast cells, membrane IgE-positive mast cells, or cytoplasmic IgE-positive mast cells have any correlation with the number of tissues eosinophils? Please describe this point in the manuscript.
2: The authors divided mast cells into membrane IgE-positive mast cells and cytoplasmic IgE-positive mast cells. Were there any mast cells which show IgE positivity for both membrane and cytoplasm? If there are, how did you classify these cells.
Author Response
Reviewer 3
The authors revealed the importance of mast cells in eosinophilic chronic rhinosinusitis. They found that the number of mast cells with membranes positive for c-kit and IgE increased significantly in lesions forming nasal polyp. In addition, they found that the number of cytoplasmic IgE-positive mast cells was significantly increased in moderate and severe ECRS. In general, manuscript is well organized and well written. There are a few changes which would improve the content.
1: Did the number of mast cells, membrane IgE-positive mast cells, or cytoplasmic IgE-positive mast cells have any correlation with the number of tissues eosinophils? Please describe this point in the manuscript.
[Reply]
Thank you for your suggestion.
In this study, we compared the number of positive cells for ECRS with the severity classification based on JESREC criteria. Eosinophil counts in tissues increased with mild ECRS (66 cells/HPF), but eosinophil counts decreased with moderate and severe ECRS (respectively 57 cells/HPF, 55 cells/HPF). In non-ECRS cases, the number of eosinophils was about 7 cells, but the number of c-kit and membrane IgE-positive cells increased to the same level as in the ECRS group. Therefore, infiltrating c-kit and IgE-positive cells did not correlate with local eosinophil counts. However, the ratio of eosinophils in blood was proportional to membrane IgE-positive mast cells and cytoplasmic IgE-positive mast cells (Figure 5C, 6I).
2: The authors divided mast cells into membrane IgE-positive mast cells and cytoplasmic IgE-positive mast cells. Were there any mast cells which show IgE positivity for both membrane and cytoplasm? If there are, how did you classify these cells.
[Reply]
Thank you for your helpful comments.
We determined mast cells that showed IgE positive only in the membrane as “membrane IgE-positive mast cells”, and mast cells that show IgE positivity in the cytoplasm ans “cytoplasmic IgE-positive mast cells”. And the number of each cell were counted.
We have modified the text (pp.7, lines7-10). The modified parts were added in blue in the text.
Round 2
Reviewer 1 Report
The paper is adequate for publication however in future studies more detailed analyses are needed to support the authors conclusions.
Reviewer 3 Report
This reviewer now agree to accept this article.
Thank you for your quick reply to my questions.
This manuscript is a resubmission of an earlier submission. The following is a list of the peer review reports and author responses from that submission.
Round 1
Reviewer 1 Report
Eosinophilic chronic rhinosinusitis (ECRS) is categorized as a subtype of chronic rhinosinusitis. In this manuscript, the authors described the association of increased mast cells in ECRS, particularly in moderate and severe ECRS. Nasal tissues from patients were analyzed with histology and IHC staining of mast cell markers c-kit and IgE. The main findings are that the number of mast cells was higher in ECRS tissues compared to Non-ECRS and controls.
Major concerns
Similar studies have been reported describing the presence of mast cells in ECRS. Particularly, Shintaro Baba et al. have published a well-performed study in the journal Annals of Allergy, Asthma & Immunology 119 (2017) 120-128.
The current work is very preliminary. There are only 4 figures. The analyses were limited to histology and IHC. Many of the important markers for mast cells, such as tryptase, cytokines were not determined.
The title is misleading. The work determined the presence of mast cells in ECRS but the presented data could not establish that mast cells caused any of the pathologies.
The literature citation was not adequate. For example, the above work by Baba et al., which was essentially the same as current work but more thorough, was not cited.
Reviewer 2 Report
In the present studies, Gion Y. et. al., investigated the presence of mast cells at the lesion sites in patients with eosinophilic chronic rhinosinusitis (ECRS) and distinct disease severities, compared to controls. The authors showed that the numbers of c-kit and IgE -expressing cells were increased in tissue containing nasal polyps as compared to uncinated tissues. They also demonstrated that the numbers of mast cells expressing cytoplasmic IgE were increased in patients with severe ECRS. Although this is an interesting study that presents certain novel data describing the presence of IgE-expressing mast cells in ECRS, there are several issues that need to be addressed to strengthen the findings and support the conclusions proposed.
Major points:
The authors should perform analyses to investigate possible correlations between mast cell numbers (as well as mast cells expressing cytoplasmic IgE) to disease parameters, such as eosinophilic inflammation (both at the lesion sites and in the peripheral blood), IgE levels in the serum, disease grading scores, etc.
Considering that inflammation and inflammatory mediators play a key role in mast cell infiltration and activation, the levels of IL-4, IL-13, IL-5 should be measured both in the serum and at the lesion sites and correlations with mast cell numbers should be presented.
The methods should clearly describe the method used for the quantification of mast cells. Were the slides studied by independent investigators, blinded to the identity of the samples? How were the cells counted? At what magnification? How many cells/what area was used for counting?
Considering that dendritic cells also express FcεR1, and therefore may express IgE, double immunohistochemical staining using a dendritic cell marker should be used to discriminate between these two cell types.
How do the authors explain that there were no differences in mast cell numbers in nasal polyp tissue between patients with and without ECRS? High magnification microphotographs showing cytoplasmic IgE-expressing mast cells in all groups should be shown. The figure legends should be more concise and contain information pertaining to the experimental approach utilized. Information pertinent to the statistical method used to analyse the data should be also provided. The Introduction should clearly present the hypothesis and aims of the present study. The Discussion should be shortened and focus only on describing the relevance of the findings. The statements presented are highly speculative and not supported by the findings.